# metaGE: Investigating genotype x environment interactions through GWAS meta-analysis

**Annaïg De Walsche**[1,2], **Alexis Vergne**[3], **Renaud Rincent**[1], **Fabrice Roux**[4], **Stéphane Nicolas**[1], **Claude Welcker**[5], **Sofiane Mezmouk**[6], **Alain Charcosset**[1], **Tristan Mary-Huard**[1,2] *

**1** Génétique Quantitative et Evolution - Le Moulon, INRAE, CNRS, AgroParisTech, Université Paris-Saclay, Gif-sur-Yvette, France, **2** MIA Paris-Saclay, INRAE, AgroParisTech, Université Paris-Saclay, Palaiseau, France, **3** CEA Tech Grand-Est, CEA, Metz, France, **4** LIPME, INRAE, CNRS, Université de Toulouse, Castanet-Tolosan, France, **5** LEPSE, Université de Montpellier, INRAE, Institut Agro, Montpellier, France, **6** KWS SAAT SE & Co. KGaA, Einbeck, Germany

* tristan.mary-huard@agroparistech.fr

**Data Availability Statement:** The datasets analysed during the current study are fully available without restriction here: https://doi.org/10.57745/

## Abstract

Elucidating the genetic components of plant genotype-by-environment interactions is of key importance in the context of increasing climatic instability, diversification of agricultural practices and pest pressure due to phytosanitary treatment limitations. The genotypic response to environmental stresses can be investigated through multi-environment trials (METs). However, genome-wide association studies (GWAS) of MET data are significantly more complex than that of single environments. In this context, we introduce `metaGE`, a flexible and computationally efficient meta-analysis approach for jointly analyzing single-environment GWAS of any MET experiment. The `metaGE` procedure accounts for the heterogeneity of quantitative trait loci (QTL) effects across the environmental conditions and allows the detection of QTL whose allelic effect variations are strongly correlated to environmental cofactors. We evaluated the performance of the proposed methodology and compared it to two competing procedures through simulations. We also applied `metaGE` to two emblematic examples: the detection of flowering QTLs whose effects are modulated by competition in Arabidopsis and the detection of yield QTLs impacted by drought stresses in maize. The procedure identified known and new QTLs, providing valuable insights into the genetic architecture of complex traits and QTL effects dependent on environmental stress conditions. The whole statistical approach is available as an R package.

## Author summary

In multi-environment trial experiments, the same panel of plants is evaluated in different well-characterized sites and years to describe the genotypic response to environmental stresses. Such experiments require dedicated statistical approaches to allow the identification of quantitative trait locus (QTLs) whose allelic effects are modulated by the stress conditions. We consider an original approach based on the joint analysis of summary

VLYYFZ. Functions to perform GWAS meta-analysis can be found in the R package metaGE, available at https://cran.r-project.org/web/packages/metaGE/index.html, along with a short tutorial on grain yield association in Maize.

**Funding:** This work was partially supported by INRAE's metaprogram DIGIT-BIO and by KWS to ADW, and partially funded by the Horizon2020 Project INCREASE, grant agreement number 862862. The funders had no role in study design, data collection and analysis, decision to publish, or preparation of the manuscript.

**Competing interests:** The authors have declared that no competing interests exist.

results from genome-wide association studies conducted separately in individual environments. Application of the method to Arabidopsis identified QTLs involved in flowering whose effects are strongly modulated by competition. Application to maize identified yield QTLs whose effects strongly correlated with the heat stress level. The method called `metaGE` drastically reduced the computational burden of the analysis and is distributed as an R package.

## Introduction

Understanding the adaptation mechanisms of plant species to different environments and the underlying genetic architecture has been a long-standing challenge in plant genetics [1–3]. It can be investigated through the mapping of quantitative trait loci (QTL) and the evaluation of QTL effects in different environments. Multi-environment trials (MET) in which the same panel of genotypes is evaluated in different locations and/or over different years have been widely adopted to dissect the genetic components underlying the genotype-by-environment (GxE) interactions and to describe the QTL response to different environmental factors. Recent genome-wide association studies in plant genetics have illustrated how the QTL–environment interaction may affect the phenotypic response [4–7]. While informative and promising, QTL identification through the analysis of MET data requires a dedicated statistical methodology to account for both the instability of allelic effects and the high level of correlation between measurements acquired on the same (or very similar) panel(s) across environments. Compared to the single environment analysis scenario for which many computationally efficient and grounded methods have been developed [8–13], the statistical analysis of MET data is still an unresolved problem with only a few methods available [14–18]. Powerful and scalable methods that account for environmental variation and its interaction with QTLs remain challenging. In this context, an appealing alternative to classical approaches consists of performing individual genome-wide association studies (GWAS) in each environment separately using one of the efficient single environment methods mentioned above, then jointly analyzing the summary results—typically the effects and p-values associated with the different markers—of the individual GWAS through a meta-analysis (MA) [19, 20].

In the context of studies conducted on different individuals, GWAS-MA has proven to yield significant gains of power over initial individual analyses while effectively controlling for false positives and keeping the computational burden low [21]. It has been successfully applied to both human [22–24] and animal [25, 26] genetic studies, allowing the detection of QTLs with small or moderate effects [27]. GWAS-MA has, however, rarely been applied to plant genetics [28, 29] and never in the context of MET analyses. In addition, most MA procedures [19, 30, 31] have been developed for human genetics purposes and aim to detect QTLs with stable effects over independent populations and, as such, are not suited to the MET context where important variations in QTL effects can be observed and where the same (or related) panel(s) is(are) phenotyped across different locations and years.

Here we introduce `metaGE`, a meta-analysis approach adapted to MET and GxE interaction analysis in plant genetics. Both fixed effect (FE) and random effect (RE) MA procedures are presented to handle fully controlled environments and experiments where the monitoring is insufficient to describe and classify the environments, as is often observed in fluctuating field conditions. To further investigate the GxE interaction, we introduce a new testing procedure to identify QTLs whose effect variations are correlated with a given environmental covariate.

We evaluated the performance and validity of the proposed methodology through detailed simulations, where `metaGE` is compared to the `METAL` meta-analysis procedure [31] currently used in most human genetics applications and the `mash` meta-analysis procedure [32] that allows the assessment of various association patterns. We further demonstrate the efficiency of `metaGE` through applications to a MET GWAS analysis of 3 species (*A. thaliana*, maize and wheat), and illustrate its versatility with an application to a maize multi-parent population MET and compare its performance to the mixed model of [33]. The whole statistical procedure is available in the `metaGE` R package.

## Materials and methods

The study of GxE interactions in plant genetics requires the evaluation of the same panel—or highly related panels including several common genotypes—in different locations and under different environmental conditions. The use of related panels in MET designs makes the independence assumption for the z-scores in the classical MA approach (see S1 Text for details) unrealistic. Depending on the experiment, environments may correspond to controlled stress conditions (*e.g.* nitrogen, water or competition stress) or to different fields and/or years where the environmental conditions are contrasted but not fully controlled by the experimenter. In this section, we show how the classical fixed and random effect MA procedures can be adapted to cope with the controlled and uncontrolled environment cases, respectively. Assuming that a GWAS analysis has been performed in each environment, the goal of the MA procedure is to summarize the environment-by-environment GWAS results while i) effectively controlling the false positive detection rate and ii) accounting for the heterogeneity of the QTL effects across environments.

We consider a meta-analysis relying on $K$ different GWASs performed in individual environments testing the association between a set of $M$ markers and a phenotype of interest. We designate $\beta_{mk}$ the estimated effect of marker $m$ in environment $k$, and $p_{mk}$ is the associated p-value. We define the z-scores $Z_{mk}$ as

$$Z_{mk} = -\Phi^{-1}(0.5p_{mk}) \times \text{sign}(\beta_{mk}),$$

where $\Phi$ stands for the standard Gaussian cumulative distribution function. The z-score is to be understood as follows: the smaller the p-value of the marker, the greater the absolute value of the z-score, with the sign of the z-score corresponding to the sign of the marker effect. Importantly, when marker $m$ is not associated with the phenotype, $Z_{mk}$ follows a standard Gaussian distribution, *i.e.* the $H_0$ distribution of $Z_{mk}$ is known. In GWAS, p-values are typically obtained by testing that no association exists between the marker under consideration and the phenotypic trait using a Wald test procedure. We point out that meta-analysis methods (including the one presented here) do not require the p-values to be obtained from a specific test procedure: it is only assumed that all p-values from the original GWAS tests are uniformly distributed over the (0,1) interval under the null hypothesis.

### Fixed effect procedure

When the environmental conditions are controlled, environments can be *a priori* classified into several groups. One can then assume the marker effect to be stable within each group but different from one group to another.

We consider that the environments are classified into $J$ distinct groups. Since the same panel of varieties—or overlapping panels—are used in all environments, the $Z$-scores cannot

be assumed to be mutually independent. The model is updated as follows:

$$Z_m = X\mu_m + E_m$$
$$E_m \sim \mathcal{N}(0_K, \Sigma_m)$$

where the incidence matrix $X$ of size $K \times J$ is such that:

$$X_{kj} = \begin{cases} 1 & \text{if the environment } k \text{ belongs to the group } j \\ 0 & \text{else} \end{cases}$$

$\mu_m = (\mu_m^1, \ldots, \mu_m^J)^T \in \mathbb{R}^J$ is the vector containing the group-specific marker effects, and $\Sigma_m$ is the inter-environment correlation matrix of size $K \times K$. If allelic effects are assumed to be stable across all environments (i.e. $J = 1$), the null hypothesis reduces to $H_0 : \{\mu_m = 0\}$ as in the FE procedure described in S1 Text.

The parameters to be estimated are the inter-environment correlation matrix $\Sigma_m$ and the marker effects within groups $(\mu_m^1, \ldots, \mu_m^J)$. We assume the inter-environment correlation matrix to be common to all markers, *i.e.* $\Sigma_m = \Sigma$. Considering only the $M_0$ markers under $H_0$, *i.e.* the markers having no effect in any environment, the correlation between the z-scores of two environments $k$ and $k'$ can be estimated as follows:

$$\hat{\Sigma}_{k,k'} = cor(Z_k, Z_{k'}) = \frac{\sum_{m=1}^{M_0}(Z_{mk} - \overline{Z_k})(Z_{mk'} - \overline{Z_{k'}})}{\sqrt{\sum_{m=1}^{M_0}(Z_{mk} - \overline{Z_k})^2}\sqrt{\sum_{m=1}^{M_0}(Z_{mk'} - \overline{Z_{k'}})^2}} \tag{1}$$

As the list of markers under $H_0$ is unknown, a filtering step for markers with a high probability of being under $H_1$ is needed. Different filtering approaches exist, and we present two of them. The first one consists of considering only the markers $m$ whose p-values $p_{mk}$ are higher than a certain threshold fixed by the user, in each environment $k \in 1, \ldots, K$ [34].

The second filtering approach that we propose consists, for each environment $k$, in estimating the distribution of the random variables $\Phi^{-1}(p_{mk})$, $m \in 1, \ldots, M$, where $\Phi^{-1}$ is the inverse distribution function of the normal distribution $\mathcal{N}(0, 1)$. By definition, the distribution to be estimated is a mixture between a $\mathcal{N}(0, 1)$ distribution corresponding to the markers under $H_0$ and a second unknown distribution corresponding to those under $H_1$. This mixture distribution can be inferred by a kernel method. Then, the filtering consists of considering only the markers $m$ whose a posteriori probabilities of being under $H_1$ are lower than a certain threshold [35]. For the applications detailed in Section Results, the filtering method was performed with a threshold set at 0.6.

Given an estimate of $\Sigma$, the within-group marker effects $\mu_m^1, \ldots, \mu_m^J$ can be inferred by the empirical means of the z-scores of each group:

$$\hat{\mu}_m = DX^T Z_m$$

where $D = \text{diag}\left(\frac{1}{n_1}, \ldots, \frac{1}{n_J}\right)$ and $n_j$ is the number of experiments in group $j$.

The association of each marker $m$ can be tested as follows:

$$H_0 : \{\forall j, \mu_m^j = 0\} \Leftrightarrow \{\mu_m = 0_J\} \quad \text{vs} \quad H_1 : \{\exists j, \ \mu_m^j \neq 0\}$$

Alternatively, one may test whether the marker $m$ has different effects across groups of environments:

$$H_0 : \{\mu_m^1 = \mu_m^2 = \cdots = \mu_m^J\} \quad \text{vs} \quad H_1 : \{\exists j_1, j_2 \ / \ \mu_m^{j_1} \neq \mu_m^{j_1}\} \tag{2}$$

More generally, one may test $H_0 : \{C\mu_m = 0\}$ for any contrast matrix $C$ of size $Q \times J$, where $Q \in \{1, \ldots, J\}$ is the number of linear constraints on the marker effects to be tested. Under the null, $C\hat{\mu}_m \underset{H_0}{\sim} \mathcal{N}(0_Q, V_C)$ where $V_C = CDX^T\Sigma XDC^T$. Therefore:

$$V_C^{-\frac{1}{2}}C\hat{\mu}_m \underset{H_0}{\sim} \mathcal{N}(0_Q, I)$$

which leads to the following test statistic:

$$T_m = \left\| V_C^{-\frac{1}{2}}CDX^TZ_m \right\|^2 \overset{H_0}{\sim} \chi^2(Q)$$

The p-value is given by:

$$p_m = 1 - \Phi_{\chi^2(Q)}(t_m),$$

with $t_m$ the observed value of $T_m$, and $\Phi_{\chi^2(Q)}$ the cumulative distribution function of the $\chi^2(Q)$ distribution.

## Random effect procedure

In the case of uncontrolled environmental conditions, the heterogeneity of the QTL effects across environments can be accounted for through a random marker effect.

The model is updated as follows:

$$\begin{aligned} Z_m &= \mu_m \mathbb{1}_K + A_m + E_m \\ E_m &= (E_{m1}, \ldots, E_{mK})^T \sim \mathcal{N}(0_K, \Sigma) \\ A_m &= (A_{m1}, \ldots, A_{mK})^T \sim \mathcal{N}(0_K, \tau_m^2 \Lambda_m) \\ E_m &\perp\!\!\!\perp A_m \end{aligned}$$

where $\tau_m^2$ and $\Lambda_m$ are the variance and the correlation matrix associated with the random marker effect, respectively.

As for matrix $\Sigma$, we assume the random effect correlation matrix to be common to all markers, *i.e.* $\Lambda_m = \Lambda$. Furthermore, since $\Sigma$ quantifies the similarity between environments, it provides some *a priori* knowledge about the similarities of allelic effects at the scale of a marker. Consequently, it will be assumed that $\Lambda = \Sigma$.

The correlation matrix $\Sigma$ is inferred using an estimator defined in Eq (1). The effect of the marker $\mu_m \in \mathbb{R}$ and the between environment variance $\tau_m^2 \in \mathbb{R}^+$ are inferred by maximum likelihood inference, yielding

$$\hat{\mu}_m = \frac{\mathbb{1}_K^T \Sigma^{-1} Z_m}{\mathbb{1}_K^T \Sigma^{-1} \mathbb{1}_K}$$

$$\hat{\tau}_m^2 = \max\left( \frac{1}{K}(Z_m - \hat{\mu}_m \mathbb{1}_K)^T \Sigma^{-1} (Z_m - \hat{\mu}_m \mathbb{1}_K) - 1, 0 \right)$$

The test for the association of the marker $m$ corresponds to:

$$H_0 : \{\mu_m = 0 \text{ and } \tau_m^2 = 0\} \quad \text{vs} \quad H_1 : \{\mu_m \neq 0 \text{ or } \tau_m^2 \neq 0\}$$

and can be performed using a likelihood ratio test. Designating $l_0$ and $l_1$ as the likelihood of $Z_m$ under $H_0$ and $H_1$, respectively, the test statistic is $2(l_1 - l_0)$ and its associated distribution under $H_0$ is a mixture of Chi-square distributions [36]:

$$\frac{1}{2}\chi^2(1) + \frac{1}{2}\chi^2(2) \ .$$

## Meta-regression test

In MET studies, trials can be characterized through some quantitative environmental covariates (*e.g.* temperature or evapotranspiration). One can then aim to identify markers whose effects are correlated to a given environmental covariate. The covariate is denoted by $X$, and $X$ is assumed to be centred. For each marker $m$, one can test:

$$H_0 : \{\mathrm{cov}(\mu_m \mathbb{1}_K + A_m, X) = 0\} \quad \text{vs} \quad H_1 : \{\mathrm{cov}(\mu_m \mathbb{1}_K + A_m, X) \neq 0\}$$

The covariance of interest can be inferred by its empirical counterpart, leading to the following test statistic:

$$\frac{Z_m^T X}{\sqrt{X^T \Sigma X}}$$

that follows a $\mathcal{N}(0, 1)$ distribution under the null hypothesis.

The p-value is given by:

$$p_m = 2 \times \Phi\left(-\left|\frac{Z_m^T X}{\sqrt{X^T \Sigma X}}\right|\right),$$

where $\Phi$ stands for the standard Gaussian cumulative distribution function.

## Multiple test control

In order to control Type I errors, we apply the local score approach developed by [37]. The local score approach detects significant regions in a genome sequence by accumulating single marker p-values while controlling the FDR. This approach depends on the choice of the threshold $\xi$ (in $\log_{10}$ scale) below which small p-values are accumulated. The significant genomic regions are computed chromosome by chromosome, and a significance threshold for the FDR control is associated with each chromosome (see [37, 38] for details). In practice, the value of $\xi$ should be selected between the average and the maximum of the set of $-\log_{10}$(p-values). In our analyses the observed average $-\log_{10}(p_m)$ ranged from 0.5 to 0.9, and the maximum value from 10 to 19, leading to a value of $\xi$ set to 3 or 4.

## Simulation framework

The performance of the proposed meta-analysis methodology was assessed through simulations. We used the genotypic data obtained from the study of [39], along with the environmental data and the experimental design described in [40]. These datasets are publicly available at https://doi.org/10.15454/AEC4BN and https://doi.org/10.15454/IASSTN, respectively.

The dataset includes 247 F1 hybrids resulting from crossing 247 diverse dent maize inbred lines with the same flint line. These hybrids were genotyped at 506,460 positions (SNPs) and phenotyped in 22 different environments. The phenotypic data were generated by simulating QTLs and a GxE genetic background. We considered four different types of QTLs:

1. **Fixed effect QTLs:** corresponds to QTL with a constant effect across all environments.

2. **Completely random effect QTLs:** corresponds to QTL whose effects are randomly drawn in each environment.

3. **Random effects based on the environment correlations QTLs:** correspond to QTL whose effects are drawn in each environment in a normal distribution whose covariance matrix reflects the similarities between environments, as measured from the environmental data.

4. **Fixed effect depending on an environmental covariate QTLs:** corresponds to QTLs whose effects are proportional to a (centred and scaled) environmental covariate.

In our simulation setting, QTL types were considered separately: 50 simulation runs were conducted for each type. In a given simulation run, phenotypic data were generated based on 12 QTLs, which were categorized into three minor allele frequency (MAF) groups: low [0.2, 0.25], medium [0.3, 0.35], and high [0.4, 0.45], with four QTLs assigned to each MAF category. These QTLs accounted for 44% of the total genetic variance, while the remaining 56% was attributed to the genetic background. The heritability in each environment was fixed at 0.5. QTL and background marker positions were randomly assigned in each run, with two chromosomes randomly designated as containing no QTLs, referred to as $H_0$ chromosomes hereafter. For each simulation run, GWAS were performed per environment using FastLMM [10]. See S2 Text for a complete description of the data simulation procedure. The resulting sets of 22 GWAS summary statistics were analyzed using the metaGE random-effect (RE) meta-analysis model for the simulations involving completely random-effect QTLs and random-effect QTLs based on environmental correlations. For simulations involving fixed-effect QTL, we applied the metaGE fixed-effect (FE) meta-analysis model. Meta-regression analyses were conducted on the three environmental covariates (Tmax, Tnight, and Psi) that were used to simulate environmental covariate-regressed QTL effects.

Additionally, the METAL procedure [31] was applied, using the FE model for fixed-effect QTL simulations and the RE model for others. The mash procedure [32] was also applied, using both canonical and data-driven covariance matrices to capture a wide range of effect patterns for the random-effect QTL and the covariate-regressed QTL simulations. In the fixed-effect QTL simulation, a specific covariance matrix effect with entries equal to one (corresponding to a shared effect for all environments) was used. When applying the mash procedure the correlation matrix was estimated from the data using a z-score threshold, as recommended in the original publication [32]. The threshold was set at 3.5, as this value resulted in a significantly reduced number of false positives for the procedure compared to the by-default threshold. For both the METAL and metaGE procedures, the local score procedure with $\xi = 4$, a nominal false discovery rate (FDR) of 0.05, and a peak threshold at 10 were used to control for multiple testing and identify putative QTL regions. As the mash procedure does not provide p-values, we followed the recommended global null test approach, considering an association significant if the local false sign rate [41] was less than 0.05 in at least one GWAS (see [32] for details). Significant regions (or markers, in the case of the mash procedure) located within 1 Mb of each other were merged into contiguous regions not exceeding 20 Mb.

False Discovery Rate (FDR) was evaluated in two different ways. A first FDR estimate (hereafter referred to as FDR_chr) was computed based on the two $H_0$ chromosomes as the percentage of simulation runs exhibiting a significant region on these chromosomes. A second FDR estimate was computed at the whole genome level as the percentage of significant local score peaks that were located outside windows of 5, 10 or 20 Mb centred on QTL positions. QTL detection power was evaluated by calculating the true positive rate, defined as the proportion of simulated QTLs correctly identified (i.e. power = Number of QTLs detected/Number of

simulated QTLs). A QTL was considered correctly detected if at least one significant peak was located within a window of 5, 10, or 20 Mb centered on the QTL position. Note that for the `metaGE` meta-regression tests, QTLs used to calculate power were limited to those associated with the corresponding covariate, i.e., four per covariate.

## Application to MET GWAS analysis

The performance of the `metaGE`, `METAL` and `mash` procedures are illustrated across four datasets selected to represent classical MET experiments for $G \times E$ interactions association studies in plant quantitative genetics. These application cases cover controlled and partially/not controlled experimental designs along with an application to multi-parent MET population. Descriptions of the datasets used in these analyses are provided below.

- **Arabidopsis dataset** [42]: GWAS analyses were performed on the local mapping population TOU-A, with a panel of 195 whole-genome sequenced accessions evaluated in six controlled micro-habitats (combinations of three soils × presence/absence of inter-specific competition, noted A to F). Each accession was phenotyped for bolting time and had genotypic information for 981,278 SNPs (after quality control and at a minor allele frequency threshold of 0.07). The trials were conducted under controlled stress conditions, with Arabidopsis being grown in competition with the weed *Poa annua* in environments B, D, F and without competition in environments A, C and E.

- **Maize dataset** [4]: GWAS analyses were performed on a panel of 244 maize dent lines evaluated as hybrids with a common parental line (a usual practice in maize genetics) in 22 environments (combinations of location × year × treatment). Each line was genotyped at 602,356 SNPs (after quality control) and phenotyped for grain yield (GY) in the 22 environments. Environmental indices were calculated as the average or the sum of one environmental variable in a period over 20 days at 20°C encompassing flowering time of the reference hybrid. The environmental variables used to calculate these indices are the soil water potential (Psi), the maximum temperature (Tmax), the night temperature (Tnight), the cumulated global radiation (Rad), the leaf to leaf Vapour Pressure Difference (VPDmax) and the Evapotranspiration (ET0). In addition, the mean night temperature during the grain filling period (Tnight.Fill) was calculated.

- **EU-NAM Flint** [43, 44]: A panel consisting of 11 biparental populations was obtained from crosses between UH007 and 11 peripheral parents representative of the Northern European maize diversity. In each population, double haploid lines were produced and genotyped at 5,263 SNPs (following quality control). All populations were evaluated for biomass dry matter yield at four locations: La Coruna, Roggenstein, Einbeck, and Ploudaniel. Three populations with fewer than 30 progenies were excluded from the present analysis.

- **Wheat dataset** [45]: GWAS analyses were performed on a panel of 210 wheat lines phenotyped for grain yield in 16 environments (combinations of location × year × treatment). Lines were genotyped at 108,410 SNPs (after quality control) and phenotyped for grain yield. This application is presented in S3 Text and S8–S10 Figs.

For the Arabidopsis, Maize, and Wheat datasets, we applied a local score approach to control for multiple testing, setting $\xi = 3$ and the nominal FDR at 0.05 for both `metaGE` and `METAL` results. Due to the lower genotypic resolution in the EU-NAM Flint dataset, which is incompatible with the local score method, we used the adaptive Benjamini-Hochberg correction procedure [46] with a nominal FDR of 0.05. The `mash` procedure was applied as in the simulation study, using either both canonical and data-driven covariance matrices to capture a

wide range of effect patterns, or a specific covariance matrix effect with entries equal to one which corresponds to a shared effect for all environments. The mash correlation matrix was estimated from the data using a z-score threshold of 3.5. The global null test was used, with a control of the local false sign rate at 0.05 [32]. All datasets used are publicly available (see the Data Availability section).

## Results

### Simulation results

Fig 1A illustrates the results obtained from the metaGE RE procedure on a single simulation run corresponding to the "completely random-effect" QTL type. The red vertical dotted lines indicate the true locations of the 12 simulated QTLs, while the blue vertical regions represent the putative QTLs identified by the procedure. Overall, the metaGE RE procedure successfully identified 7 of the 12 QTLs within a 5 Mb detection window. Fig 1B focuses on chromosome 3, which illustrates the characteristic behaviour of the local score procedure. The region containing the simulated QTL was initially tagged by ten significant local score peaks, along with several smaller residual peaks. After the merging and thresholding steps, the number of peaks was reduced to three. The peak falling within the 5 Mb QTL detection window is indicated by the red horizontal line at the bottom of the figure. The two remaining peaks, labelled as "FP" (false positives) on the graph, were classified as false positives for the 5 Mb window and as true positives when evaluated against detection windows of 10 Mb and 20 Mb, respectively. Among the QTLs undetected by the metaGE RE procedure, three corresponded to QTLs with a low MAF between 0.20 and 0.25, and two corresponded to QTLs with a medium MAF between 0.30 and 0.35. On this same dataset, the mash procedure [32] identified 65 significant regions after merging the significant SNPs located within 1 Mb. Of these, 53 were outside the 5 Mb QTL detection windows, yielding a false discovery rate of 0.81. The remaining 13 regions fell within the detection windows of 10 QTLs. The METAL procedure [31] resulted in 14 significant local score peaks after the merging and thresholding steps, all being outside the 5 Mb QTL detection windows. These results are representative of the ones obtained for all the simulation runs, regardless of the simulated QTL type.

Table 1 displays the FDR_chr and the average FDR and power for each meta-analysis procedure across different QTL types, evaluated with 5 Mb, 10 Mb, and 20 Mb detection windows. The FDR_chr was lower than 0.05 for the different metaGE MA procedures. For the two

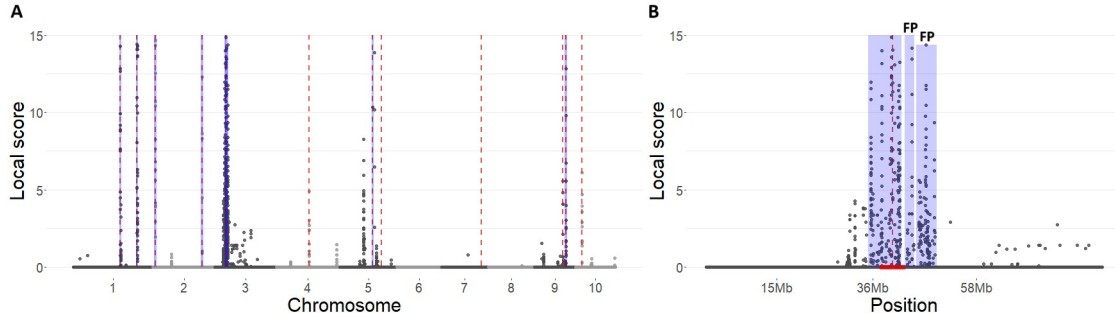

**Fig 1. Results of the metaGE RE procedure applied to a simulation run corresponding to the "completely random-effect" QTL type.** (A) Local scores along the chromosomes. The red vertical dotted lines correspond to the localization of the 12 simulated QTLs. The range of values on the y-axis has been bounded from 0–15 to highlight minor QTL. The blue boxes represent the significant zones identified. (B) Same representation restricted to a QTL region on chromosome 3. The red horizontal bottom line corresponds to the 5Mb QTL detection windows. FP stands for False Positive.

**Table 1. Performance over 50 simulation runs for each simulation scenario, using a 5, 10 or 20 Mb detection window size.**

| Method | QTL_type | FDR_chr | FDR_5Mb | FDR_10Mb | FDR_20Mb | Power_5Mb | Power_10Mb | Power_20Mb |
|---|---|---|---|---|---|---|---|---|
| METAL_FE | Fixed | 1 (0) | 0.93 (0.01) | 0.9 (0.01) | 0.86 (0.02) | 0.98 (0.03) | 0.99 (0.03) | 1 (0.02) |
| METAL_RE | Random | 1 (0) | 0.94 (0.04) | 0.91 (0.05) | 0.84 (0.07) | 0.16 (0.13) | 0.22 (0.16) | 0.29 (0.17) |
| METAL_RE | RandomCov | 1 (0) | 0.93 (0.02) | 0.9 (0.03) | 0.84 (0.04) | 0.58 (0.2) | 0.64 (0.21) | 0.76 (0.19) |
| METAL_RE | Reg | 0.88 (0.33) | 0.94 (0.15) | 0.9 (0.17) | 0.86 (0.17) | 0.06 (0.07) | 0.1 (0.1) | 0.15 (0.13) |
| mash_FE | Fixed | 0.14 (0.35) | 0.32 (0.33) | 0.28 (0.31) | 0.23 (0.28) | 0.2 (0.15) | 0.21 (0.16) | 0.22 (0.16) |
| mash | Random | 0.78 (0.42) | 0.8 (0.08) | 0.74 (0.09) | 0.66 (0.11) | 0.79 (0.14) | 0.8 (0.14) | 0.82 (0.13) |
| mash | RandomCov | 0.88 (0.33) | 0.84 (0.06) | 0.79 (0.07) | 0.7 (0.09) | 0.53 (0.14) | 0.56 (0.14) | 0.59 (0.14) |
| mash | Reg | 0.86 (0.35) | 0.85 (0.06) | 0.81 (0.08) | 0.74 (0.1) | 0.41 (0.17) | 0.44 (0.17) | 0.49 (0.18) |
| metaGE_FE | Fixed | 0 (0) | 0.1 (0.27) | 0.1 (0.27) | 0.09 (0.26) | 0.09 (0.08) | 0.09 (0.08) | 0.09 (0.08) |
| metaGE_RE | Random | 0.04 (0.2) | 0.1 (0.15) | 0.07 (0.12) | 0.05 (0.1) | 0.51 (0.14) | 0.51 (0.14) | 0.51 (0.14) |
| metaGE_RE | RandomCov | 0.02 (0.14) | 0.18 (0.2) | 0.13 (0.19) | 0.08 (0.16) | 0.26 (0.11) | 0.26 (0.11) | 0.27 (0.11) |
| metaGE_MR_Psi | Reg | 0.02 (0.14) | 0.09 (0.18) | 0.09 (0.18) | 0.08 (0.17) | 0.52 (0.25) | 0.52 (0.25) | 0.52 (0.26) |
| metaGE_MR_Tmax | Reg | 0 (0) | 0.1 (0.22) | 0.08 (0.19) | 0.04 (0.13) | 0.26 (0.23) | 0.26 (0.23) | 0.27 (0.23) |
| metaGE_MR_Tnight | Reg | 0 (0) | 0.12 (0.23) | 0.08 (0.2) | 0.06 (0.18) | 0.34 (0.24) | 0.34 (0.24) | 0.34 (0.24) |

FDR_chr refers to the percentage of simulation runs exhibiting a significant region on the two $H_0$ chromosomes. FDR_XMb refers to the percentage of significant local score peaks located outside windows of X Mb centered on QTL positions. Power_XMb is the proportion of simulated QTLs correctly identified by at least one significant peak within a window of X Mb centered on the QTL position. Numbers in brackets correspond to standard errors.

RE = Random effect procedure; FE = Fixed effect procedure; MR_covariate = meta-regression test associated with the covariate; mash_FE = the mash procedure using only the matrix with all ones as covariance matrix; QTL type Fixed = Fixed effect QTLs; QTL type Random = Completely random effect QTLs; QTL type RandomCov = Random effects based on the environment correlations QTLs; QTL type Reg = Fixed effect depending on an environmental covariate QTLs; Tnight = the night temperature; Tmax = the maximum temperature; Psi = the soil water potential.

competing methods, the FDR_chr was systematically higher than the nominal level, ranging from 0.14 to 0.88 for mash and close to 1 in most scenarios for METAL.

At the whole genome level, the FDR was high, ranging from 0.1 to 0.18 for metaGE to more than 0.93 for METAL when using the most stringent detection window size (5Mb). The whole genome FDR decreased with respect to the detection window size, as expected, but remained above 0.84 for METAL and 0.23 for mash even when considering the largest detection window (20Mb).

Detection powers were around 0.09, 0.39 and 0.37 for the metaGE FE model, RE model and meta-regression tests, respectively. Overall, both METAL and mash procedures displayed higher detection power than metaGE across most QTL types, but their prohibitive False Positive levels undermined the reliability of the detected associations. Note that despite its higher False Positive rate, METAL achieved the lowest detection power for the completely random-effect QTLs and QTLs regressed on environmental covariates. One could also observe a strong impact of the Minor Allele Frequency (MAF) on the detection power for the metaGE procedure, as illustrated in Table 2. Specifically, the detection power for QTLs with high MAF was three times greater than that for those with low MAF (e.g. from 0.04 to 0.14 in the case of metaGE_FE).

Although the detection power performance of the meta-regression (MR) tests reported in Table 1 only accounts for QTLs specifically correlated to the corresponding covariate, it may happen that a meta-regression test detects QTLs associated with another covariate that the one under study. Table 3 displays the QTLs detected by the 3 MR tests. The MR test associated with the Tnight covariate detected 34.5% of its target QTLs over the 50 simulation runs. It also detected 5.5% of the Tmax target QTLs but did not detect any of the Psi target QTLs. This can be directly related to the correlations between the different environmental covariates: the

**Table 2. Average detection power for the different minor allele frequency levels over the 50 simulation runs, using a 5 detection window size.**

| Method | QTL_type | Power_Low_MAF | Power_Medium_MAF | Power_High_MAF |
|---|---|---|---|---|
| metaGE_FE | Fixed | 0.04 (0.1) | 0.08 (0.15) | 0.14 (0.18) |
| metaGE_RE | Random | 0.36 (0.23) | 0.54 (0.26) | 0.62 (0.24) |
| metaGE_RE | RandomCov | 0.15 (0.2) | 0.28 (0.18) | 0.36 (0.22) |
| metaGE_MR_Psi | Reg | 0.4 (0.44) | 0.52 (0.44) | 0.54 (0.43) |
| metaGE_MR_Tmax | Reg | 0.13 (0.3) | 0.31 (0.4) | 0.31 (0.42) |
| metaGE_MR_Tnight | Reg | 0.16 (0.3) | 0.33 (0.41) | 0.51 (0.45) |

Each simulation included four QTLs of each MAF levels. Numbers in brackets correspond to standard errors.

RE = Random effect procedure; FE = Fixed effect procedure; MR_covariate = meta-regression test associated with the covariate; Tnight = the night temperature; Tmax = the maximum temperature; Psi = the soil water potential; QTL type Fixed = Fixed effect QTLs; QTL type Random = Completely random effect QTLs; QTL type RandomCov = Random effects based on the environment correlations QTLs; QTL type Reg = Fixed effect depending on an environmental covariate QTLs; Minor allele frequency (MAF) levels: low [0.2, 0.25], medium [0.3, 0.35], and high [0.4, 0.45].

**Table 3. Percentage of QTLs detected by each `metaGE` meta-regression test across 50 simulation runs.**

| Meta-regression test | QTL_Tnight | QTL_Tmax | QTL_Psi |
|---|---|---|---|
| metaGE_MR_Tnight | 34.5 | 5.5 | 0 |
| metaGE_MR_Tmax | 7 | 26.5 | 0 |
| metaGE_MR_Psi | 0 | 0.5 | 52 |

Each simulation included four QTLs with fixed effects dependent on each of the three covariates. MR_covariate = meta-regression test associated with the covariate; QTL_covariate = QTL with fixed effect depending on the covariate; Tnight = the night temperature; Tmax = the maximum temperature; Psi = the soil water potential.

correlation between Tnight and Tmax was 0.71, whereas the correlation between Tnight and Psi was -0.11.

## Genetic response to competition in Arabidopsis

We consider the Arabidopsis dataset from [42], which includes six GWAS analyses aimed at identifying associations with bolting time across six controlled micro-habitats, labelled A through F. Inter-specific competition was present in environments B, D, and F, while environments A, C, and E were competition-free.

Three different meta-analysis procedures were used to analyze the six GWAS summary statistics: the `metaGE` FE procedure, the `mash` procedure [32] using only the matrix with all ones as covariance matrix, corresponding to a shared effect for all environments, and the `METAL` procedure. Fig 2 shows the resulting histograms and Q-Q plots of the p-value distributions from the `METAL` procedure and the `metaGE` FE method for quality control purposes. The `METAL` q-q plot clearly exhibits an inflated proportion of low p-values, with more than 165,000 p-values under 0.01, compared to the $\approx$10,000 that one would expect under the full $H_0$ hypothesis. This inflated proportion results in declaring more than 15% of the markers as associated after an FDR correction. In comparison, the `metaGE` FE procedure effectively controls the type I error rate, as demonstrated by the p-value distribution. It led to the identification of 191 SNPs clustered into 61 QTLs. The `mash` procedure identified 401 SNPs, with the corresponding peaks mostly overlapping the ones identified with the `metaGE` procedure (S1 Fig).

Out of the 61 putative QTLs that were identified through the `metaGE` FE procedure, 51 were significant in at least one individual GWAS (see S3 Text for details). Following the methods of [47], the list of significant SNPs identified by the `metaGE` method was further

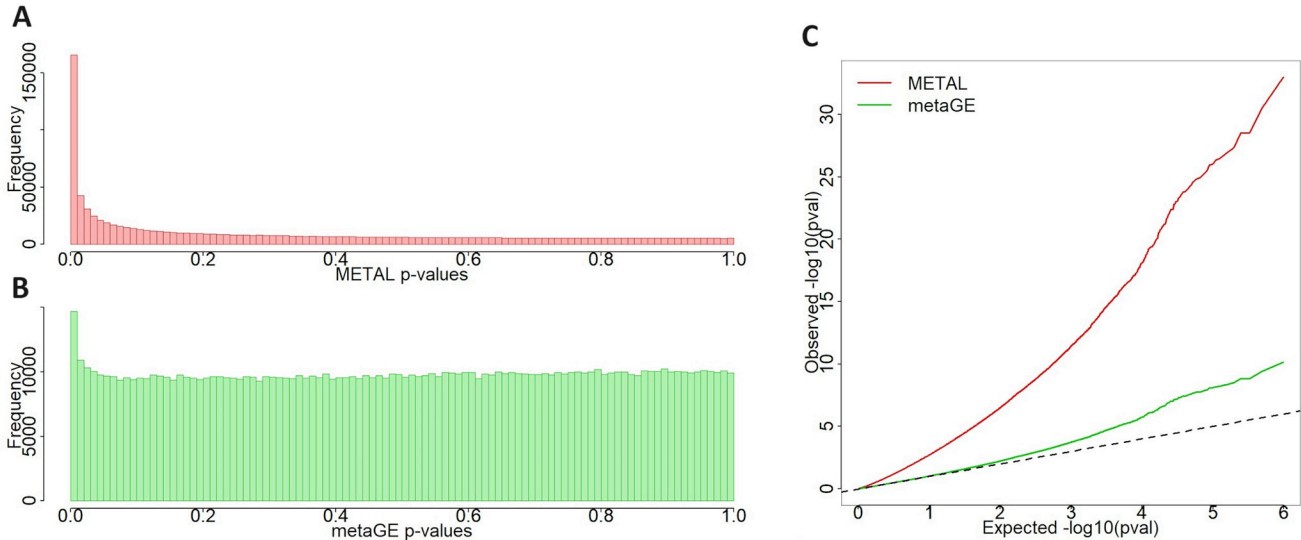

**Fig 2. P-value distributions of `METAL` and `metaGE` FE procedures applied to Arabidopsis dataset.** (A) Histogram of the `METAL` p-values. (B) Histogram of the `metaGE` FE p-values. (C) QQ-plot of the -log10(p-values) of `METAL` in red and `metaGE` in green. The observed -log10(p-values) are compared to the expected quantiles generated by the uniform null distribution.

corroborated by examining the enrichment ratio for *a priori* candidate genes linked to bolting time. The markers identified by the `metaGE` approach were found to have a significantly higher enrichment (enrichment = 4.13) than what would be expected at random ($[q_{0.05}; q_{0.95}]$ = [0.066; 3.2]).

We then applied the `metaGE` contrasted FE testing procedure described in the Methods section, Eq (2) with a group effect corresponding to the presence or absence of competition with *Poa annua* to detect markers with contrasted allelic effects based on the presence or absence of competition. The `metaGE` contrasted FE procedure identified 221 SNPs located in 72 QTLs (Fig 3A) and covering 160 candidate genes that showed significant enrichment for three Gene Ontology terms (MapMan functional annotation, [48]), *i.e.* 'development' (P = 8.866e-03), 'cell' (P = 1.522e-03) and 'tetrapyrrole synthesis' (P = 0.020). Interestingly, the two latter MapMan processes were also detected as enriched when subjecting the local

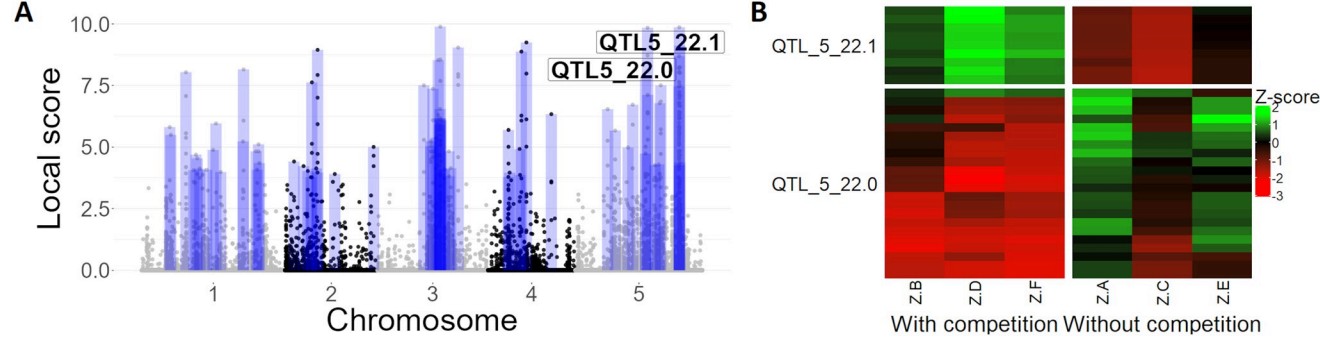

**Fig 3. Results of the `metaGE` FE procedure applied to the Arabidopsis dataset to detect markers with competition contrasted effects.** (A) Local scores along the chromosomes. The blue boxes represent the significant zones identified. (B) Z-scores of two QTL regions located on chromosome 5 (*QTL5_22.0* a,d *QTL5_22.1*), with markers in rows and environments in columns. A, C and E correspond to the 3 environments without competition.

mapping population TOU-A to the presence of three weed species in greenhouse conditions [49].

Fig 3B represents the z-scores of two top QTLs found on chromosome 5 involving 22 and 9 markers, respectively (local score of 8.69 and 9.1). The identified markers clearly exhibited a contrasted marker effect profile: when the effects were positive in one of the two groups of environments, they were negative or null in the other. All the 22 SNPs of the second QTL (*QTL5_22.1*) are located in the AtCNGC4 genomic region that is well-known to affect floral transition [50, 51].

Importantly, as the `metaGE` contrasted FE procedure aimed to identify markers with contrasted (*i.e.* unstable) effects across the two sets of environments, 71 out of the 72 QTLs identified by the contrasted FE procedure were new candidates that were not detected by the standard FE procedure.

### Genetic response to drought in maize

We consider the Maize dataset of [4], where GWAS analyses were performed to identify association with grain yield in 22 environments (combinations of location × year × treatment).

The `metaGE` RE procedure detailed in Materials and methods section was applied to perform the joint analysis of the 22 per environment GWAS summary statistics. In total, 52 genomic regions were identified, of which 14 correspond to QTLs also detected in the original publication [4]. The three putative QTLs with the most significant association peaks (Fig 4A) were located on chromosomes 3 (*QTL3_120.0*) (local score = 38), 6 (*QTL6_20.3*) (local score = 415) and 7 (*QTL7_41.4*) (local score = 18, not detected in the original publication). The z-score heatmaps corresponding to *QTL6_20.3* and *QTL7_41.4* are displayed in Fig 4B. The heatmap of *QTL6_20.3* revealed a cluster of six environments (Cra12R, Cam12R, Cra12W, Mur13R, Mur13W and Cam12W) with strong similar marker effects that were found to be characterized by severe heatwaves with high night temperatures (close to 22˚C) and high maximal temperatures (above 36˚C) together with high evaporative demand during the day (3.6 KPa). The *QTL6_20.3* individual GWAS p-values were all lower than 1e-6 in the six aforementioned environments, confirming the strong association signal detected by the RE procedure. In contrast, the *QTL7_41.4* showed a moderate positive effect across nearly half of the environments—individual GWAS p-values were lower than 1e-2 in ten environments out of 22—and was found to be significant in only two individual GWAS.

On the same dataset, `METAL` detected 40% of the markers as significant (S2 Fig). The results of the `mash` procedure were similar to the ones obtained with `metaGE`, except for chromosome 9 where `mash` declared 909 markers covering most of the chromosome (from 13.4Mb to 154.4Mb) as significant (S3 Fig). See S3 Text for details.

We performed a genome-wide detection of markers whose effect variations were correlated with environmental variables by running the meta-regression test procedure described in Materials and methods section. The environmental variables considered were the evapotranspiration, the mean night temperature during the flowering period (Tnight) and the mean night temperature during the grain filling period (Tnight.Fill).

Regarding evapotranspiration, the meta-regression test identified 14 QTLs located on chromosomes 2 and 9 (Fig 4C). Allelic effects on grain yield of one of the most significant markers (marker AX-91538480, *QTL2_153.8*) substantially changed with potential evapotranspiration (Fig 4D).

For the mean night temperature during the flowering period, the meta-regression test identified 21 QTLs, with the main association corresponding to a genomic region located at less

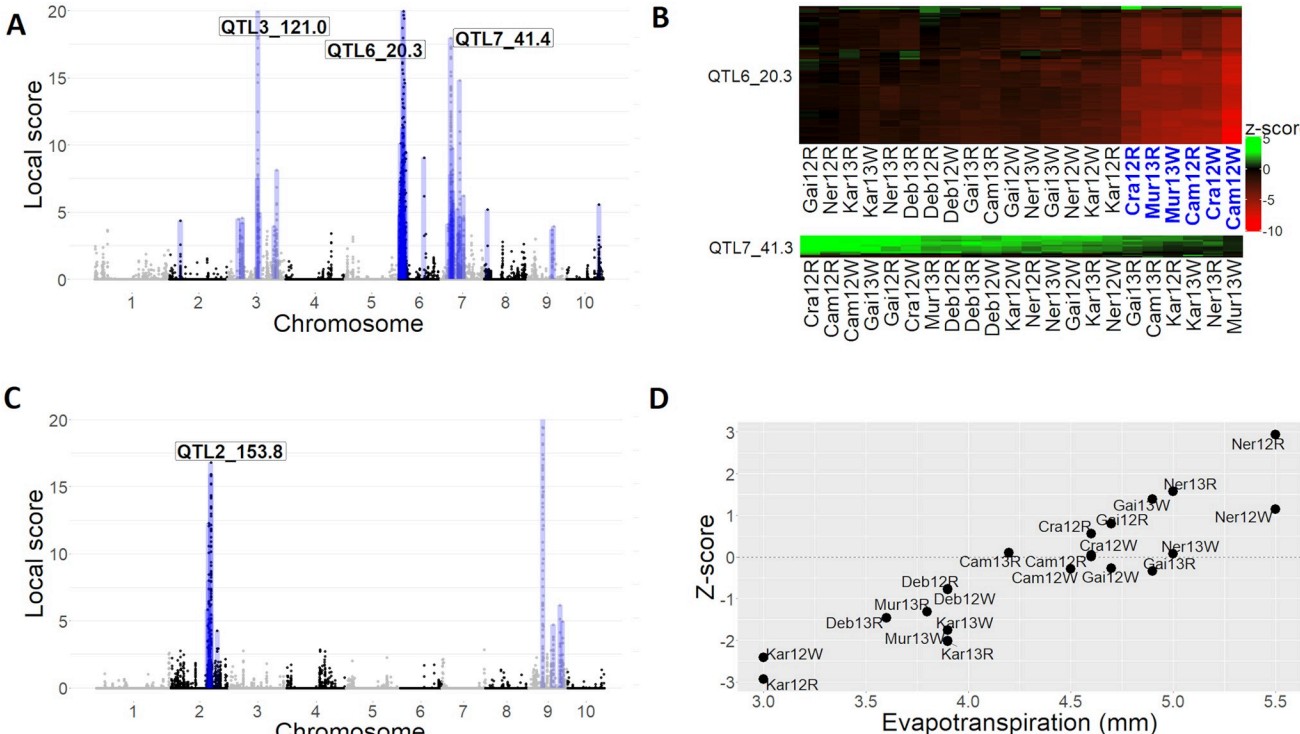

**Fig 4. Results of the `metaGE` procedures applied to the Maize dataset.** (A) Local score obtained from the `metaGE` RE procedure along the chromosomes. The range of values on the y-axis has been bounded from 0–20 to highlight minor QTLs. The blue boxes represent the significant zones identified. (B) Z-scores of two QTLs located on chromosomes 6 and 7 (*QTL6_20.3* and *QTL7_41.4*), with markers in rows and environments in columns. Only the 100 top significant SNPs out of 169 composing the *QTL6_20.3* have been displayed. The six environments highlighted in blue correspond to environments characterized by severe heatwaves (night temperature close to 22°C and maximal temperature above 36°C). (C) Local score obtained from the meta-regression test for the evapotranspiration along the chromosomes. The range of values on the y-axis has been bounded from 0–20 to highlight minor QTL. The blue boxes represent the significant zones identified. (D) Z-scores as a function of the evapotranspiration, for the top significant marker of *QTL2_162.5* detected with the meta-regression procedure (Maize dataset, marker AX-91538480).

than 0.6 Mb from *QTL6_20.3* (S4(A) Fig). This finding corroborates the results previously obtained in [4].

Regarding the night temperature during the grain filling period, the meta-regression test identified 15 QTLs across six chromosomes. In particular, the allelic effects on GY of one of the most significant markers located on chromosome 9 (marker AX-91123283, *QTL9_28.6*) changed dramatically according to night temperature during the grain filling period, with positive effects on cool nights and negative effects on hot nights during the grain filling period (S4(B) Fig). See S3 Text for an analysis regarding night temperatures during the grain filling period.

## Extension to Multi-Parental Population MET experiments

Unlike previous datasets involving an association panel, the present dataset consists of a multi-parent crossing design, with each bi-parental progeny being phenotyped in four locations (La Coruna, Roggenstein, Einbeck and Ploudaniel). In this context, we extend the notion of environment to the combination of one sub-population and one location.

To estimate the allelic effect per parent and environment, GWAS analyses were performed on each combination of cross and location, resulting in 32 individual analyses. The `metaGE`

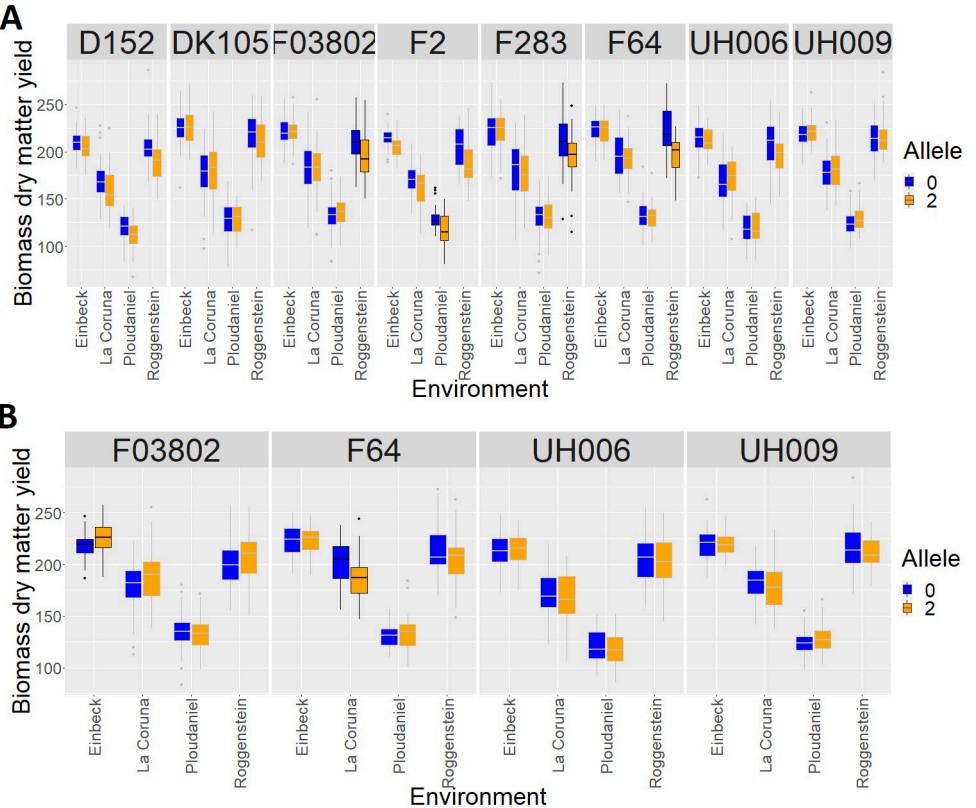

**Fig 5. Results of the `metaGE` RE procedure applied to the EU-NAM Flint dataset.** (A) Allelic effects of *QTL6_84.2* (EU-NAM Flint, marker PZE.106101278) through locations and sub-populations (B) Allelic effects of *QTL5_23.9* (EU-NAM Flint, marker PZE.105012387) through locations and sub-populations. A combination of location × sub-population is highlighted where individual GWAS p-values were below 0.01.

RE procedure detailed in Materials and methods section was applied to perform the joint analysis of the 32 individual GWAS summary statistics. In total, 16 QTLs were identified, highlighting some very significant association peaks, especially on chromosomes 1 (*QTL1_117.6*) and 6 (*QTL6_84.2*). These two QTLs were also identified in the publication of Garin et al. [33]. The allelic effect of *QTL1_117.6* was almost consistent across all populations except in F2 (S7 Fig). *QTL6_84.2* showed an interesting genetic effect series since an ancestral allele inherited by parents D152, F03802, F2, F283, UH006, and DK105 [33] had a strong negative effect, primarily in the environment TUM (Fig 5A).

While the analysis of Garin et al. [33] was limited to the study of two of the four locations, the present analysis included all locations. In this study, we revealed ten new QTLs not detected in the initial analysis, five of which corresponded to QTLs, which showed effect inversion among populations. For example, *QTL5_23.9* showed a positive effect for the population F03802 and a negative effect for the population F64 (Fig 5B). We also noticed that 3 of them were associated with the flowering time QTL detected from the same materials in Giraud et al. [52]. Flowering time has a simpler genetic determinism than grain yield and is one of its main drivers, with negative, null or positive correlations according to environmental conditions [4]. The EU-NAM Flint dataset was also analyzed with the `METAL` and `mash` procedures. Similar to the previous application cases, `METAL` led to an inflated detection rate ($\approx$ 30% of the

**Table 4. Computational time for the meta-analysis procedures across different datasets.**

| Dataset | Nb. Env. | Nb. Markers | Method | Time |
|---|---|---|---|---|
| Simulation | 22 | ≈ 500K | METAL_FE | 1.1min |
| | | | METAL_RE | 2.6min |
| | | | mash | 16.6min |
| | | | mash_FE | 20s |
| | | | metaGE_FE | 38s (30s) |
| | | | metaGE_RE | 49s (31s) |
| | | | metaGE_MR | 51s (30s) |
| Arabidopsis | 6 | ≈ 1M | METAL_FE | 2.6min |
| | | | mash_FE | 29s |
| | | | metaGE_FE | 1.2min (26s) |
| Maize | 22 | ≈ 600K | METAL_RE | 3.3min |
| | | | mash | 25.3min |
| | | | metaGE_RE + MR | 2.25min (41s) |
| EU-NAM Flint | 32 | ≈ 6, 000 | METAL_RE | 3s |
| | | | mash | 1.8min |
| | | | metaGE_RE | 12s (8s) |
| Wheat | 16 | ≈ 100K | METAL_RE | 22s |
| | | | mash | 3.3min |
| | | | metaGE_RE | 47s (30s) |

Numbers in brackets correspond to the time of the inference of the inter-environment correlation matrix. For the simulated datasets, the computational time were averaged over the simulation runs.

RE = Random effect procedure; FE = Fixed effect procedure; MR = meta-regression test; mash_FE = the `mash` procedure using only the matrix with all ones as covariance matrix.

markers, S5 Fig) while `mash` detected 278 markers, including the ones corresponding to the three main peaks identified by `metaGE`, S6 Fig. See S3 Text for details.

## Computational efficiency

Table 4 displays the computational times for the `metaGE`, `METAL` and `mash` procedures in simulated data and real datasets. In the simulation studies, all procedures run the analysis in less than 3 minutes, except for the `mash` procedure using canonical and data-driven covariance matrices that required 16.6 minutes.

The analysis of the different real datasets was handled in less than 3 minutes using the `metaGE` procedure, even for datasets characterized by a large number of environments and/ or markers. Note that for `metaGE`, most of the required computational time corresponds to the inference of the inter-environment correlation matrix, a step that needs to be run only once as it is common to both the RE and FE models and all subsequent testing procedures.

## Discussion

We demonstrated the potential of the `metaGE` meta-analysis procedure to address the analysis of MET GWAS analysis through a simulation study and four case studies on *A. thaliana*, maize, and also wheat (see S3 Text), covering a diversity of experimental designs and trait complexities. The `metaGE` method addresses a need in the community, as current meta-analysis procedures do not combine the different features required for a full exploration of

MET analysis. We also introduced some new key extensions for investigating the relationship between QTL effects and some qualitative or quantitative covariates.

## Comparison to existing methods

Hereafter, we provide a typology of methodological requirements that are key for MET analysis and, more generally, GxE applications. We discuss how our developments to address these are connected to the many MA procedures developed in the last decade motivated by other fields of application, mainly in the context of human genetics.

A by-default feature of most popular MA procedures [31, 53] is the assumption of independence between GWAS. However, as shown by the behaviour of METAL in our simulation studies and real data applications, ignoring dependencies when applying MA on MET analysis significantly inflates the Type I error rate, with FDR exceeding 0.84 across all simulated scenarios. We hypothesize that this assumption of independence is a key reason why MA has seen limited application in MET and GxE analyses. Some recent works have extended the standard MA model to account for dependency between GWAS results. In [54], the authors proposed a method to handle potential overlapping samples by computing correlations based on information on shared subjects. In the context of pleiotropy analysis, where multiple traits measured on the same panel are jointly analyzed, different strategies have been proposed to infer correlations between summary statistics, either based on LD-score regression [55–57] or directly from the z-scores [32, 34, 58, 59]. The metaGE procedure used to estimate the correlation matrix between environments aligns with these approaches. It can be related to the inference procedure of the correlation matrix between traits described in [32, 34] but with a different strategy for filtering $H_1$ markers. In those methods, the filtering is achieved by directly applying thresholds to p-values or z-scores. In contrast, the metaGE filtering approach relies on the posterior probabilities of markers being under $H_1$ as obtained from the procedure detailed in [35].

Most GxE experiments are characterized by complex designs that aim to compare two or more environmental scenarios. These scenarios may include the presence/absence of competitors (e.g. adventices) or pathogens, abiotic stress (e.g. drought or nitrogen stress), or contrasted climatic conditions (Temperature gradient). Environments may then be *a priori* sorted into groups according to the environmental scenario, with the aim of comparing the marker effect stability between groups. However, standard meta-analysis methods like METAL assume that marker effects are consistent across all GWAS studies, which complicates the detection of unstable or environment-dependent markers, as shown in our simulations. In contrast, the mash method can model different effect patterns but cannot account for patterns influenced by environmental factors, which is essential for studying genotype-environment interactions. Alternatively, a wide range of MA procedures have been developed in contexts other than association genetics to jointly analyze summary statistics from independent studies [60]. In particular, subgroup MA or tests of contrast [61, 62] and meta-regression [63, 64] can be used to explore the relationship between the summary statistics and a study-level categorial or quantitative covariate, respectively. Recently, in [65], the subgroup MA was applied to gestational diabetes mellitus association studies to investigate the relationship between different confounding factors or demographic characteristics of the study populations and the marker effect estimates. Additionally, [66] developed a MA of independent GWAS results that explores all possible subsets of association studies and identifies the subset with the strongest association signal. However, to our knowledge, these MA procedures were not available in the context of non-independent studies.

## GWAS-MA confers gain in power over individual GWAS

GWAS-MA has proven to yield significant gains in power over individual preliminary analyzes while effectively controlling for false positives [21]. The observed gain in detection power applies to the MET context as well. The `metaGE` MA approach revealed interesting new QTLs even in regions where the test statistics did not pass the nominal significance threshold in most individual environments. For example, the genomic region corresponding to the third highest local score in the Maize dataset (*QTL7_41.4*, local score = 18) was found to be significant in only two environments out of 22 and was not detected in the original study [4]. This region harbours QTLs controlling plant growth rate and final biomass in water deficit conditions [67]. Similarly, in the Wheat dataset, none of the QTLs identified by the `metaGE` RE method were found to be significant in any environment. Therefore, the joint analysis of individual GWAS is suitable for complex traits (such as yield) whose genetic variations are usually due to many QTLs with minor effects that might go undetected in single-environment analyses.

## Interpreting variability in QTL effects across environmental conditions

The MA extensions proposed here facilitate the assessment of the stability or variation in QTL effects across environmental conditions, which is of key interest in detecting alleles conferring specific adaptive features. As an illustration, the analysis of the Arabidopsis dataset highlighted a major region on chromosome 5 (*QTL5_22.0*), with an effect sign that switched according to the presence or absence of competition with *Poa annua*. Arabidopsis *QTL5.220* is located in the AtCNGC4 genomic region that is well-known to affect floral transition [50, 51]. In addition, AtCNGC4 impairs plant immunity [51, 68], which is in line with the negative effect of competitive interactions on pathogen defense to the benefit of plant development, such as with floral transition [69].

Similar results were found in the maize analysis, where environmental conditions were not controlled a priori. The analysis of allelic effects of *QTL6_20.3* detected with the RE model highlighted a group of six environments (Fig 4C). A subsequent analysis showed that these environments were characterized by severe heat waves at night. Consistently, *QTL6_20.3* colocalized with a QTL affecting grain yield in the "Po valley" in which maize is currently irrigated but subjected to high temperatures during summer [70]. *QTL6_20.3* also overlaps with a large 2.4 Mbp present/absent variant harbouring dozens of genes—including one encoding an ABA-induced protein in response to water deficit [4]. These genes were further shown to be associated with environmental adaptation to high temperatures and to have undergone strong selection during both domestication and improvement [71].

Beyond *a posteriori* interpretation, environmental variables can also be incorporated as a covariate in the GWAS-MA model for a whole genome scan of the response of QTL effects. Such a relationship between marker effects and environmental variables was investigated in [4] for the Maize dataset. However, this initial analysis was restricted to markers found to be significantly associated in at least one environment due to the prohibitive computational time required to perform a genome-wide analysis. To address this limitation, we proposed a meta-regression test designed to identify QTLs whose effect variations correlate with a specified environmental covariate. The performance of the meta-regression test was evaluated through a simulation study, which demonstrated proper FDR control and high detection power. The simulation study also pointed out that when two environmental covariates are highly correlated, the meta-regression test applied to the first covariate may detect QTLs whose effect variations are linked to the second covariate. Although this finding does not impact the performance of the procedure, it emphasizes that careful interpretation is necessary in cases of high correlation between environmental covariates.

We performed the genome-wide analysis of the Maize dataset with the `metaGE` meta-regression procedure using three covariates: evapotranspiration, mean night temperature during the flowering and grain filling periods. This analysis led to the identification of 14, 20, and 14 new regions, respectively. Fig 4D illustrates how the effect of a QTL located on chromosome 2 (pos154) varies linearly from negative to positive effects according to evapotranspiration. This QTL colocalizes with the QTL of expression of aquaporins (eQTL of PIP2.2 ez and eQTL of PIP2.1 ez). These channel proteins facilitate water transport between cells and impact water use efficiency (WUE) and stomatal conductance [72]. These two physiological adaptive traits are highly sensitive to environmental conditions [67]. A second region was detected on chromosome 9 (pos125) using the same environmental covariate. It colocalizes with a QTL for plant growth (PG) rate and a QTL for WUE under water deficit conditions. Both PG and WUE are traits that are highly sensitive to evaporative demand. Lastly, a third QTL, also located on chromosome 9 (pos135), colocalizes with a QTL of plant growth sensitivity to soil water potential, consistent with previously observed effects [67]. None of these genomic regions on chromosomes 2 and 9 were detected in the initial publication.

## Benefits of working with summary statistics

An attractive feature of MA procedures is their ability to cope with unbalanced/incomplete data—without the need for further data imputation or additional computational overhead. The procedures presented in this article rely on summary statistics (*i.e.* p-values and effect signs) obtained from per-environment GWAS, which enables the avoidance of re-scaling if traits were measured on different scales. The use of summary statistics also makes the addition/removal of a given environment (based *e.g.* on *post hoc* quality control) and the update of the results straightforward.

Regarding the genotypic information, MA procedures can be used to easily handle cases where summary statistics are missing for some markers in a series of environments. This may occur *e.g.* when different technologies or sequencing depths were used in the individual experiments. Furthermore, in plant genetics, multi-parental population analyses are pervasive and also yield missing summary statistics, as different sets of markers may be monomorphic in different parental subpopulations. Our method is still applicable in such cases, as illustrated with the EU-NAM example, where inversions of allelic effects were identified for some subpopulations (Fig 5). These inversions may correspond to a genetic background effect (*i.e.* to conditional epistasis, see for instance [73]), or alternatively to an allelic series with several haplotypes presenting a range of effects, with the SNP allele contrasting central haplotypes with extreme opposite haplotypes.

In human genetics, meta-analysis methods have facilitated collaborations and data sharing between different consortia while preserving individual data confidentiality, as only summary statistics are required for the integrated association analysis. As an illustration, the recent Global Biobank Meta-analysis Initiative [74] gathered data from 24 BioBanks covering 2.2 million research participants, resulting in large-scale genetic association studies with improved detection power. We strongly believe that the meta-analysis procedures presented in this paper will open the way to similar consortia in plant breeding by allowing private companies to share GWAS results without the need to share individual plant genotypic or phenotypic information.

## GWAS-MA is an efficient and versatile tool for GxE analysis

While several approaches have been proposed to model the GxE interactions in different contexts [14–17, 33], many of them require an amount of computational time that may become prohibitive when dealing with either large scale genomic datasets or a large number of

environments. In contrast, the MA procedure introduced here is computationally efficient, processing large datasets, such as 22 environments and approximately 600,000 markers, in a few minutes (Table 4).

In addition, the `metaGE` procedures can be applied to various MET designs such as controlled experiments, uncontrolled or partially controlled experiments and all kinds of multi-parent population METs widely used in modern breeding programs. Our methodology integrates several tools for the characterization of environments. The contrast test associated with the FE procedure enables the identification of QTLs specific to some subsets of environments, *i.e.*, whose behaviour interacts with the environmental characterization defined by the classification. The classification definition is very flexible and can rely on any qualitative covariate, such as known control/stress conditions or sub-populations in the case of an MPP experiment. The meta-regression test allows the detection of QTL, whose variations are correlated to any quantitative environmental covariate. Searching for such relationships between marker effects and environmental characteristics is a key issue in plant genetics. It has also been widely investigated in the context of animal and human MA studies, where dedicated procedures have been developed to handle environmental covariates measured at the individual level [75, 76]. Our meta-regression procedure represents a new contribution to account for environmental covariates measured at the environment level and complements the existing MA toolbox. All the testing procedures presented here, along with the FE and RE procedures, are implemented in the `metaGE` R package available on the CRAN repository.

In recent years, a number of public initiatives involving thousands of individuals and evaluated in dozens of well-characterized environments have been developed, such as the Genomes To Fields project for maize [77] or the elite yield trial nurseries from CIMMYT's bread wheat breeding program [78, 79]. METs are one of the central elements of a breeding program [80]. Screening such datasets for genome-wide associations requires the development of scalable and flexible methodological tools. In this context, GxE Meta-Analysis can be routinely applied in most breeding programs, and we strongly believe that it will represent a methodology of choice in the future to address the many challenges of modern GWAS.

## Supporting information

**S1 Text. Description of meta-analysis classical approach.**
(PDF)

**S2 Text. Details of the simulation framework.**
(PDF)

**S3 Text. Additional analysis of real data applications.**
(PDF)

**S1 Fig. Results of the `metaGE` FE and the `mash` procedures applied to the Arabidopsis dataset.** (A) Local scores obtained from the `metaGE` FE procedure along the chromosomes. The blue boxes represent the significant zones identified. The range of values on the y-axis has been bounded from 0–15 to highlight minor QTLs. (B) Minimum local false sign rate over the environments obtained from the `mash` procedure along the chromosomes (in $\log_{10}$ scale). The horizontal red line represents the significance threshold of 0.05.
(TIF)

**S2 Fig. P-value distributions of the `METAL` and the `metaGE` RE procedures applied to Maize dataset.** (A) Histogram of the `METAL` p-values. (B) Histogram of the `metaGE` RE p-values. (C) QQ-plot of the -log10(p-values) of `METAL` in red and `metaGE` in green. The

observed -log10(p-values) are compared to the expected quantiles generated by the uniform null distribution.
(TIF)

**S3 Fig. Results of the `mash` procedure applied to the Maize dataset.** (A) Minimum local false sign rate over the environments obtained from the `mash` procedure along the chromosomes (in $\log_{10}$ scale). The range of values on the y-axis has been bounded from 0–10 to highlight minor QTLs. The horizontal red line represents the significance threshold of 0.05. The vertical dotted lines correspond to the three main regions identified by the `metaGE` RE procedure located on chromosomes 3 (*QTL3_120.0*), 6 (*QTL6_20.3*) and 7 (*QTL7_41.4*). (B) Z-scores of the significant SNPs located on chromosome 9, with markers in rows and environments in columns.
(TIF)

**S4 Fig. Night temperature meta-regression tests applied to the Maize dataset.** (A) Z-scores as a function of the mean night temperature during the flowering period for the top significant marker detected with the meta-regression procedure (Marker AX-91369217 located on chromosome 6 pos 21Mb). (B) Z-scores as a function of the mean night temperature during the grain filling period for the top significant marker detected with the meta-regression procedure (Marker AX-91123283 located on chromosome 9 pos 28Mb).
(TIF)

**S5 Fig. P-value distributions of the `METAL` and the `metaGE` RE procedures applied to EU-NAM Flint dataset.** (A) Histogram of the `METAL` p-values. (B) Histogram of the `metaGE` RE p-values. (C) QQ-plot of the -log10(p-values) of `METAL` in red and `metaGE` in green. The observed -log10(p-values) are compared to the expected quantiles generated by the uniform null distribution.
(TIF)

**S6 Fig. Results of the `mash` and the `metaGE` RE procedures applied to the EU-NAM Flint dataset.** (A) P-values of the random-effect `metaGE` procedure along the chromosomes (in $-\log_{10}$ scale). The horizontal red line represents the significance threshold of the adaptive Benjamini Hochberg multiple testing correction procedure [46] for a nominal FDR of 0.05. (B) Minimum local false sign rate over the environments obtained from the `mash` procedure along the chromosomes (in $-\log_{10}$ scale). The horizontal red line represents the significance threshold of 0.05.
(TIF)

**S7 Fig. Allelic effects of QTL1_117.6 (marker PZE.101144585) identified by the `metaGE` RE procedure applied to the EU-NAM Flint dataset.** A combination of location x sub-population is highlighted if individual GWAS p-values were below 0.01.
(TIF)

**S8 Fig. Results of the `metaGE` procedures applied to the Wheat dataset.** (A) Local score along the chromosomes from the `metaGE` RE procedure. The boxes represent the significant zones identified. (B) Local score along the chromosomes from the `metaGE` FE procedure. The boxes represent the significant zones identified. (C) Z-scores as a function of the correlation between the heading date and the grain yield, for the top significant marker detected with the meta-regression procedure (Wheat dataset, marker cfn2941229 located on chromosome 6A). (D) Z-scores as a function of the correlation between the heading date and the grain yield, for the second top significant marker detected with the meta-regression procedure (Wheat dataset, marker cfn1693678 located on chromosome 2B). Colours correspond to the environment

classification according to the relationship between heading date and grain yield of [45].
(TIF)

**S9 Fig. P-value distributions of the `METAL` and `metaGE` RE procedures applied to Wheat dataset.** (A) Histogram of the `METAL` p-values. (B) Histogram of the `metaGE` RE p-values. (C) QQ-plot of the -log10(p-values) of `METAL` in red and `metaGE` RE in green. The observed -log10(p-values) are compared to the expected quantiles generated by the uniform null distribution.
(TIF)

**S10 Fig. Results of the `mash` procedures applied to the Wheat dataset.** Minimum local false sign rate over the environments obtained from the `mash` procedure along the chromosomes (in $-\log_{10}$ scale). The horizontal red line represents the significance threshold of 0.05. The blue dots correspond to the SNPs indentified by the `metaGE` RE procedure.
(TIF)

## Author Contributions

**Conceptualization:** Tristan Mary-Huard.

**Data curation:** Annaïg De Walsche, Renaud Rincent, Fabrice Roux, Stéphane Nicolas, Claude Welcker.

**Formal analysis:** Annaïg De Walsche, Alain Charcosset, Tristan Mary-Huard.

**Funding acquisition:** Sofiane Mezmouk, Tristan Mary-Huard.

**Methodology:** Annaïg De Walsche, Alexis Vergne, Tristan Mary-Huard.

**Project administration:** Sofiane Mezmouk, Tristan Mary-Huard.

**Software:** Annaïg De Walsche, Alexis Vergne, Tristan Mary-Huard.

**Supervision:** Alain Charcosset, Tristan Mary-Huard.

**Writing – original draft:** Annaïg De Walsche, Renaud Rincent, Fabrice Roux, Stéphane Nicolas, Claude Welcker, Alain Charcosset, Tristan Mary-Huard.

**Writing – review & editing:** Annaïg De Walsche, Alain Charcosset, Tristan Mary-Huard.

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
