## [Decision Letter · Decision Letter 0]

19 Sep 2024

Dear Dr Mary-Huard,

Thank you very much for submitting your Research Article entitled 'metaGE: Investigating genotype x environment interactions through GWAS meta-analysis' to PLOS Genetics.

The manuscript was fully evaluated at the editorial level and by independent peer reviewers. The reviewers appreciated the attention to an important topic but identified some concerns that we ask you address in a revised manuscript.

We therefore ask you to modify the manuscript according to the review recommendations. Your revisions should address the specific points made by each reviewer.

To resubmit, log into your Editorial Manager account and select the option 'Revise Submission' in the 'Submissions Needing Revision' folder.

Yours sincerely,

Angela Hancock, Ph.D.

Academic Editor

PLOS Genetics

Michael Epstein

Section Editor

PLOS Genetics

Your manuscript has now been reviewed by two expert reviewers. Both were generally positive, but they also had some questions and suggestions to clarify points made in the manuscript. In your revised submission, please respond to all comments and questions from the reviewers.

Reviewer's Responses to Questions

**Comments to the Authors:**

Reviewer #1: In this manuscript, De Walsche et al. present metaGE, an R package for the analysis of correlated marker effects in GWAS meta-analyses. The manuscript is generally very well-written. The authors rigorously describe the statistical procedures implemented in their tool, and they apply it to several datasets which illustrate its statistical and practical advantages. Remarkably, the metaGE package can be easily downloaded from CRAN, and contains useful help pages (in which example code can be run smoothly) and a vignette. My main comments are related to the simulation studies and the description of empirical studies. The manuscript would be clearer with more controlled simulation studies (see my comment below), a systematic description of analyses (e.g., regarding significance thresholds), and consistent comparisons (e.g., between metaGE and METAL whenever relevant).

Major comment:

Material and Methods, Simulation framework: How did you choose QTLs in simulations, and why did you choose to simulate different types of QTLs (FE, RE, Correlated RE, proportional to environmental covariate) together in the same simulation? The simulation results would be much easier to interpret (by the authors) and understand (by the readers) if factors impacting QTL detection power were controlled: allele frequency of QTLs, and type of QTL effect. Currently, it's difficult to make sense of the simulation results because of the heterogeneity in marker effects.

Minor comments:

Material and Methods: Please add a subsection about the publicly available datasets that you use: species, number of lines and SNPs, number of environments and environmental covariates, phenotypic traits, and significance threshold used for significance in each dataset. This information should be available in the main text.

Results: Why did you not compare metaGE with METAL in the analysis of maize and wheat datasets? I suggest you consistently provide a description of results from METAL, for comparison.

Results, Computational efficiency of the method: How do the computational times compare with competing methods like METAL?

L. 68: Does the p-value need to be produced by a specific type of test? Here, does it need to come from a Wald test? Please mention this information in this paragraph.

L. 83: Please specify that Sigma_m is a K x K matrix.

L. 86: Please use a subcript J for the 0-vector.

L. 131: Was there a typo in the formula for the estimated variance over environments? Shouldn't it be (1/(K-1)) * crossproduct, instead of (1/K) * crossproduct - 1?

L. 134: Could you provide a reference for the test with a mixture of chi-squares, as I don't think it's common knowledge in the quantitative genetics community: for example, case 6 in

Self, S. G., & Liang, K.-Y. (1987). Asymptotic properties of maximum likelihood estimators and likelihood ratio tests under nonstandard conditions. Journal of the American Statistical Association, 82(398), 605–610. https://doi.org/10.1080/01621459.1987.10478472

L. 140-144: I suggest you use upper-case Z and X (as random variables) when describing estimators, not lower-case z and upper-case X, which is a bit confusing – especially since X is assumed to be fixed for the calculation of the SE of Z_m^TX in L.141.

L. 145-153: What is the justification for using epsilon = 3? Please provide an explanation, or even a sensitivity analysis, which would be useful since Fariello et al. (ref. 35) recommend epsilon = 1 or 2.

L. 157-159: the links https://doi.org/10.15454/AEC4BN and https://doi.org/10.15454/IASSTN were not working at the time of my review.

L. 190-195: How did you choose the distance thresholds for merging detected QTLs (1 Mb) and the detection window size (5, 10, or 20 Mb)? Do these choices depend on the rate of LD decay in the population under study?

Table 1: Please add results from METAL, for comparison with metaGE.

Table 2: Please add a legend to describe the numbers in this table: metareg.X and QTL.X, total number of runs per test and number of simulated QTL per run.

L. 293: "supported" or "corroborated" instead of "validated"?

L. 428-429: Please elaborate on the different strategies for filtering H1 markers.

L. 488-489: How many new regions did you identify?

L. 493: "In prep." citations are generally not acceptable. Can't you cite other studies to support your claim about these aquaporin genes?

Reviewer #2: The work by Walsche et al Investigated GxE in GWAS meta-analysis using simulated data and four case studies of A. thaliana, maize, and also wheat. Authors introduce a powerful R package that run the analysis in a user-friendly way. However, some comments are listed below to improve the manuscript.

- Not clear to me the concept of the MA, does the analysis run on all SNPs or only on the significant ones?

- What are the advantages of the procedure introduced here over other approaches such as "mash" or even the ones implemented in GenStat? Both are used in plant research. This point should be added to the discussion.

- Is the metaGE analysis time consuming like GenStat, when using large genotypic data. That is not clear here but can be a good advantage.

Some sentences in the results should be moved to the discussion part, for example, lines 234-244: "This can be explained by the fact that these procedures have less targets than the RE procedure, e.g. the FE procedure aims at identifying QTLs with stable allelic effects, i.e. the QTL of type 1 in our simulations." Please, pay attention for this point all over the results.

**Have all data underlying the figures and results presented in the manuscript been provided?**

Reviewer #1: Yes

Reviewer #2: Yes

PLOS authors have the option to publish the peer review history of their article (what does this mean?). If published, this will include your full peer review and any attached files.

Reviewer #1: No

Reviewer #2: No

---

## [Decision Letter · Decision Letter 1]

10 Dec 2024

PGENETICS-D-24-00868R1

metaGE: Investigating genotype x environment interactions through GWAS meta-analysis

PLOS Genetics

Dear Dr. Mary-Huard,

Thank you for submitting your manuscript to PLOS Genetics. After careful consideration, we feel that it has merit but does not fully meet PLOS Genetics's publication criteria as it currently stands. Therefore, we invite you to submit a revised version of the manuscript that addresses the points raised during the review process.

Please submit your revised manuscript within 30 days Jan 09 2025 11:59PM. If you will need more time than this to complete your revisions, please reply to this message or contact the journal office at plosgenetics@plos.org. Please include the following items when submitting your revised manuscript:

We look forward to receiving your revised manuscript.

Kind regards,

Angela Hancock, Ph.D.

Academic Editor

PLOS Genetics

Michael Epstein

Section Editor

PLOS Genetics

Aimée Dudley

Editor-in-Chief

PLOS Genetics

Anne Goriely

Editor-in-Chief

PLOS Genetics

**Additional Editor Comments:**

We have now received reviews of the revised version of your manuscript from the expert reviewers who reviewed the first version. The reviewers mentioned a few minor issues that should be addressed for the final version. Please address all comments from the reviewers. In particular, there was a question about some references to the 'FDR' were actually referring to false positive (FP) rate rather than FDR = FP/(FP+TP). If this is true, please make sure this is clear in the manuscript. The second reviewer asked about some seeming discrepancies in Table 4. Please make sure this information is consistent and clear in the table and provide an explanation in the response to reviewers.

Thank you and we look forward to the final version.

**Journal Requirements:**

1) Please upload a copy of Figures 2, 3, 4, and and 5. which you refer to in your text on page. Or, if the figure is no longer to be included as part of the submission please remove all reference to it within the text.

2) Please ensure that the funders and grant numbers match between the Financial Disclosure field and the Funding Information tab in your submission form. Note that the funders must be provided in the same order in both places as well.

**Reviewers' comments:**

Reviewer's Responses to Questions

**Comments to the Authors:**

Reviewer #1: De Walsche et al. addressed all my previous comments very well. They made genuine efforts to improve the simulation studies, add one highly relevant competing method (mash) in their analyses, and conduct systematic comparisons with competing methods. Specifically, the comparisons of statistical performances (Table 1), computational efficiencies (Table 4) and functionalities (Discussion section) are great additions to their study. As a result of their thorough revision, the quality of their article is even higher. I only have very minor comments (see below).

Figures 2 to 5 were missing from this new submission. I assumed that they were the same as those from the previous submission. Please confirm.

Table 1: please describe power and FDR in the legend. These are worth defining, especially FDR which here refers to the false positive rate, and not the estimated FDR of detected effects (e.g., by the Benjamini-Hochberg procedure).

Table 3: Please mention/remind (e.g., in the title) that the MR test is part of metaGE.

L. 463, Table 4: ‘min’ instead of ‘mn’

L. 528: ‘effectively’ instead of ‘efficiently’

L. 628: Refer to Table 4 at the end of this sentence.

Reviewer #2: The authors have addressed all my concerns, and the article now reads more comprehensive.

I just want to draw authors' attention to a minor comment that came to my mind regarding Table 4 and the speed of each procedure. How could Mash take 40 minutes to analyze 500 K SNPs over 22 environments for simulated data, but faster with Maiz data with 600 K SNPs and 22 environments? Same for metaGE_RE, not clear on which factor is time consuming, number of environments or markers. Is there any conclusion whether the speed is related with number of markers or environments?

Regarding GenStat, I used 214 K SNPs with 260 accession over few environments, and it took 2-3 days to get the analysis done. metaGE is really powerful.

**Have all data underlying the figures and results presented in the manuscript been provided?**

Reviewer #1: Yes

Reviewer #2: Yes

PLOS authors have the option to publish the peer review history of their article (what does this mean?). If published, this will include your full peer review and any attached files.

Reviewer #1: No

Reviewer #2: **Yes: **Mohamed El-Soda

**Figure resubmission:**
---

## [Editor Report · Decision Letter 2]

23 Dec 2024

Dear Dr Mary-Huard,

We are pleased to inform you that your manuscript entitled "metaGE: Investigating genotype x environment interactions through GWAS meta-analysis" has been editorially accepted for publication in PLOS Genetics. Congratulations!

Yours sincerely,

Angela Hancock, Ph.D.

Academic Editor

PLOS Genetics

Michael Epstein

Section Editor

PLOS Genetics

Aimée Dudley

Editor-in-Chief

PLOS Genetics

Anne Goriely

Editor-in-Chief

PLOS Genetics

Comments from the reviewers (if applicable):

The authors have addressed the remaining minor issues raised by reviewers in their revised manuscript. I recommend that the manuscript now be accepted.

**Data Deposition**

http://datadryad.org/submit?journalID=pgenetics&manu=PGENETICS-D-24-00868R2

**Press Queries**

---

## [Editor Report · Acceptance letter]

2 Jan 2025

PGENETICS-D-24-00868R2 

metaGE: Investigating genotype x environment interactions through GWAS meta-analysis 

Dear Dr Mary-Huard, 

We are pleased to inform you that your manuscript entitled "metaGE: Investigating genotype x environment interactions through GWAS meta-analysis" has been formally accepted for publication in PLOS Genetics! Your manuscript is now with our production department and you will be notified of the publication date in due course.

With kind regards,

Zsofia Freund

PLOS Genetics

On behalf of:
